# Cell-free prototyping enables implementation of optimized reverse β-oxidation pathways in heterotrophic and autotrophic bacteria

Bastian Vögeli [1], Luca Schulz [1], Shivani Garg[2], Katia Tarasava[3], James M. Clomburg[2,3], Seung Hwan Lee [3], Aislinn Gonnot[2], Elamar Hakim Moully[4,5], Blaise R. Kimmel [1], Loan Tran[2], Hunter Zeleznik[2], Steven D. Brown[2], Sean D. Simpson[2], Milan Mrksich [1,4,5], Ashty S. Karim [1], Ramon Gonzalez [3✉], Michael Köpke [2✉] & Michael C. Jewett [1✉]

Carbon-negative synthesis of biochemical products has the potential to mitigate global $CO_2$ emissions. An attractive route to do this is the reverse β-oxidation (r-BOX) pathway coupled to the Wood-Ljungdahl pathway. Here, we optimize and implement r-BOX for the synthesis of C4-C6 acids and alcohols. With a high-throughput in vitro prototyping workflow, we screen 762 unique pathway combinations using cell-free extracts tailored for r-BOX to identify enzyme sets for enhanced product selectivity. Implementation of these pathways into *Escherichia coli* generates designer strains for the selective production of butanoic acid ($4.9 \pm 0.1$ gL$^{-1}$), as well as hexanoic acid ($3.06 \pm 0.03$ gL$^{-1}$) and 1-hexanol ($1.0 \pm 0.1$ gL$^{-1}$) at the best performance reported to date in this bacterium. We also generate *Clostridium autoethanogenum* strains able to produce 1-hexanol from syngas, achieving a titer of 0.26 gL$^{-1}$ in a 1.5 L continuous fermentation. Our strategy enables optimization of r-BOX derived products for biomanufacturing and industrial biotechnology.

---

[1] Department of Chemical and Biological Engineering and Center for Synthetic Biology, Northwestern University, Evanston, IL, USA. [2] LanzaTech Inc., Skokie, IL, USA. [3] Department of Chemical, Biological, and Materials Engineering, University of South Florida, Tampa, FL, USA. [4] Department of Biomedical Engineering, Northwestern University, Evanston, IL, USA. [5] Department of Chemistry, Northwestern University, Evanston, IL, USA. ✉email: ramongonzale@usf.edu; michael.koepke@lanzatech.com; m-jewett@northwestern.edu

Current extractive industrialization processes annually release 9 gigatons (Gt) of $CO_2$ (total anthropogenic $CO_2$ emission >24 Gt), while only fixing approximately 120 megatons[1]. This considerable imbalance in the global carbon cycle is a leading cause of climate change, motivating the need for both new $CO_2$ waste gas recycling strategies and carbon, as well as energy-efficient routes to chemicals and materials[2]. Industrial biotechnology is one promising carbon-recycling approach to address this need. For example, *Clostridium autoethanogenum* has emerged as a cellular factory to convert carbon oxides in the atmosphere (e.g., CO, $CO_2$) and green hydrogen into sustainable products, like ethanol[3,4]. Unfortunately, designing, building, and optimizing biosynthetic pathways in cellular factories remains a complex and formidable challenge.

A key issue is that past metabolic engineering efforts have chiefly focused on linear heterologous pathways that limit the co-development of multiple biochemical products. Cyclic and iterative pathways offer the unique advantage of providing access to hundreds of molecules with different chemistries and chain lengths from one core pathway. Reverse β-oxidation (r-BOX) is one such cyclic pathway that is highly modular and has been demonstrated as a promising route to many sustainable molecules with different chemistries and carbon chain length[5,6]. For example, r-BOX has been successfully used in *Escherichia coli* to produce high titers of C4-C10 saturated[7] and α,β-unsaturated carboxylic acids[8] (Fig. 1A), adipic acid, or tiglic acid[9]. In addition to *E. coli*, r-BOX has been shown to work in a range of platforms that offer process advantages and can

access alternative feedstocks[5,10]. While promising for manufacturing diverse products from a single iterative pathway, r-BOX studies have mostly shown the production of biochemical mixtures. This necessitates extensive downstream purification[11] and presents a major challenge of r-BOX in engineering and controlling product selectivity. Selection of unique combinations of enzyme homologs and expression levels is required to achieve product selectivity, adding development time, cost, and optimization bottlenecks.

We have recently established a cell-free framework for in vitro prototyping and rapid optimization of biosynthetic enzymes (iPROBE)[12]. This workflow is able to accelerate the testing of biochemical pathways from months-to-years in non-model organisms (e.g., *C. autoethanogenum*), or weeks in model organisms (e.g., *E. coli*), to just a few days, while also increasing the number of pathways that can be tested[12]. The platform uses combinatorial assembly of enzyme homologs produced in vitro using cell extract and DNA templates to study and tune pathway performance. For example, iPROBE allowed us to screen 54 different linear pathways for 3-hydroxybutyrate production, 205 variants of a butanol pathway, and 580 unique linear pathway combinations for limonene synthesis starting with glucose as a carbon source[13–17]. iPROBE additionally enabled the optimization of 15 competing reactions for acetone biosynthesis in *C. autoethanogenum*[18] and pathway performance in each study correlated well with in vivo yields. Adapting iPROBE to study cyclic and iterative pathways like r-BOX could allow engineering product specificity but has not previously been attempted.

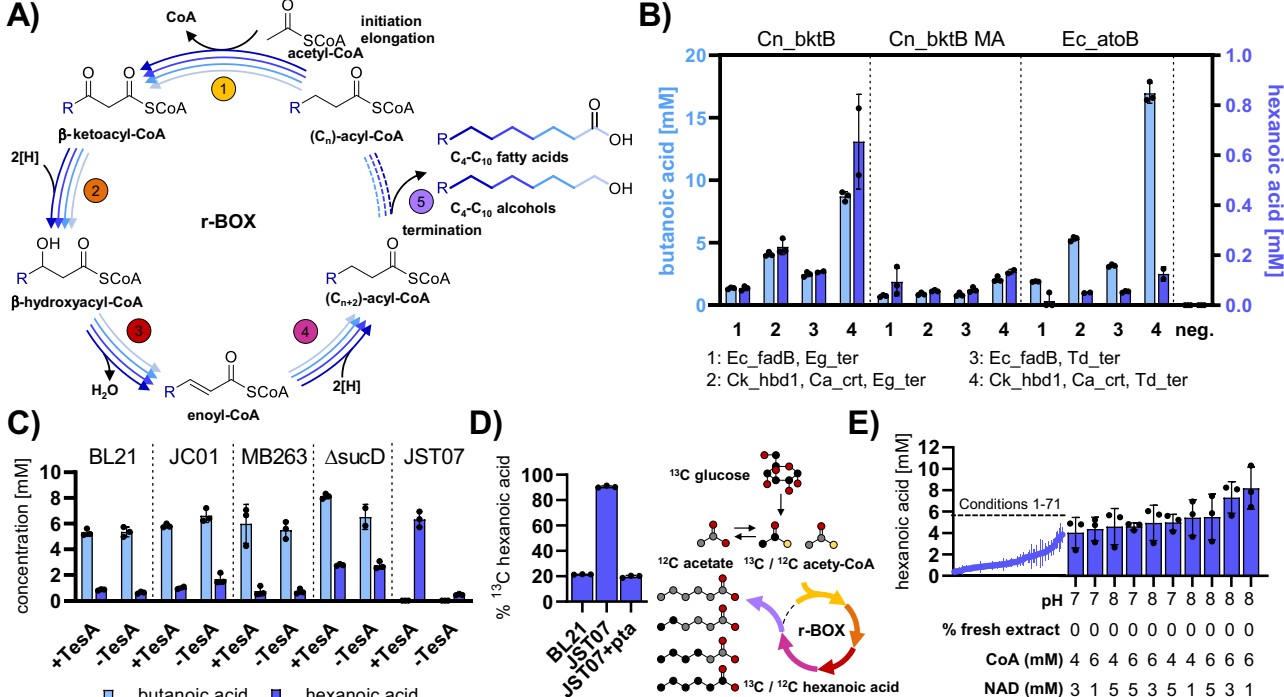

**Fig. 1 Establishing a cell-free r-BOX platform. A** General scheme of r-BOX. A thiolase (1) adds an acetyl-CoA unit every cycle to a growing acyl-CoA chain. The 3-keto group is then successively reduced (2), dehydrated (3) and reduced (4) again to form a two-carbon elongated acyl-CoA group. Termination enzymes (5) generate products from the acetyl-CoA intermediates of this iterative process. **B** Initial screen of r-BOX enzymes to establish a functional base case for in vitro hexanoic acid synthesis. All assays contained 0.3 μM of each cycle enzyme and 0.15 μM Ec_tesA. Conditions (i) Ec_fadB, Eg_ter; (ii) Ck_hbd1, Ca_crt, Eg_ter; (iii) Ec_fadB, Td_ter; (iv) Ck_hbd1, Ca_crt, Td_ter. The highest hexanoic acid production was observed for Cn_bktB, Ck_hbd1, Ca_crt, Td_ter, which was used in all future optimizations. **C** Thioesterase background activity from the extract strain used for iPROBE was assessed by omitting thioesterase Ec_tesA. **D** Carbon utilization assays using the base case of r-BOX enzymes with 120 mM $^{13}$C glucose. The assay contains approximately 120 mM $^{12}$C acetate from extract preparation and addition as CFME salts. % of $^{13}$C label in the hexanoic acid product was determined by GC-MS. $^{13}$C labeled carbons are shown in black, $^{12}$C labeled carbons in gray. **E** Buffer, concentration of glycolytic enzymes, and cofactor optimization for the initial r-BOX set. Detailed results and full information on enzyme abbreviations in Source Data file. Data represent $n = 3$ independent experiments, with the standard deviation and mean shown.

Here, we adapt iPROBE to optimize r-BOX for the specific production of butanol, butanoic acid, hexanol, and hexanoic acid. In addition, we develop an automated liquid-handling workflow, allowing us to screen 440 unique enzyme combinations and 322 assay conditions. A key feature of the workflow was the application of high-throughput characterization of CoA metabolite concentrations in reactions containing variable sets of r-BOX enzymes. We use self-assembled monolayers for matrix-assisted laser desorption/ionization-mass spectrometry (SAMDI-MS) for this analysis.[17] We show that the source strain used for iPROBE extracts influences the yields and specificity of the screened pathways and use previous in vivo optimizations in *E. coli* to find an optimal strain for r-BOX cell-free prototyping[9,11]. We identify specific pathways for all four target products and demonstrate—for the first time—that pathway performance correlates well across three platforms: a cell-free system, a heterotrophic model organism (*E. coli*), and an autotrophic organism (*C. autoethanogenum*) capable of using syngas as the sole carbon and energy source. We were able to implement our optimized r-BOX pathways and generate strains with the highest to date reported titers of $3.06 \pm 0.03$ gL$^{-1}$ hexanoic acid and $1.0 \pm 0.1$ gL$^{-1}$ hexanol in *E. coli*, and 0.26 gL$^{-1}$ hexanol in *C. autoethanogenum*. The direct correlation between prototyping in vitro and in vivo implementation in two metabolically different organisms forms a new blueprint for the generation and optimization of biochemical pathways for metabolic engineering and synthetic biology.

## Results

**r-BOX—a model system for optimization in vitro, in *E.coli* and in *C. autoethanogenum*.** The modular and cyclic nature of r-BOX provides access to hundreds of possible molecules, but tuning product selectivity remains a key challenge. To address this issue, we developed and used high-throughput cell-free prototyping via iPROBE to tailor product selectivity and enable in vivo implementation (Supplementary Fig. 1). Specifically, we targeted medium chain (C4-C6) fatty acids and alcohols due to the potential to produce them at high yield and the lack of green chemistry approaches. We established the r-BOX pathway in a cell-free system, extensively screened pathway enzymes via iPROBE, and explored pathway flux via SAMDI-MS to inform pathway design in living systems. Notably, selecting r-BOX, which has been well-studied in the model organism *E. coli*[5,7,9,11], gave us a sufficient starting point to benchmark in vitro prototyping and allowed for the direct comparison of r-BOX pathway performance between an in vitro system, *E. coli*, and *C. autoethanogenum*.

**Strain selection and automation of r-BOX with the iPROBE framework.** To establish r-BOX in the cell-free environment, we produced the four enzymes required for initiation (thiolase – TL) and elongation of product chain-length (hydroxyacyl-CoA dehydrogenase – HBD, crotonase – CRT, trans-enoyl-CoA reductase – TER), along with a termination enzyme required for carboxylic acid production (thioesterase – TE, Fig. 1A) using cell-free protein synthesis (CFPS). All selected enzymes and the abbreviations used throughout this manuscript are listed in Source Data file – Enzyme list. We initially picked 12 enzyme sets chosen from successful C4-C10 carboxylic acid biosynthesis in *E. coli*[19] and in vitro production of butanol[12] (Fig. 1B). Enzyme homologs were expressed using the PANOx-SP CFPS system in *E. coli* crude cell lysates[20], which contain all the endogenous glycolysis enzymes to convert glucose to acetyl-CoA—the starting substrate of r-BOX—while regenerating NADH[21]. The resultant enzyme-enriched extracts were mixed in ratios to assemble r-BOX and ensure 0.3 μM final concentrations of each pathway enzyme, which has been shown to be sufficient for prototyping similar metabolic pathways in iPROBE[12]. The cell-free mixture was then incubated with salts, buffers, glucose as a carbon source, and catalytic amounts of NAD$^+$ at 30 °C for 24 h. The best initial enzyme combination for hexanoic acid produced $0.65 \pm 0.20$ mM hexanoic acid, but also produced $8.7 \pm 0.5$ mM butanoic acid as an early termination product (Fig. 1B). Previous knockout studies in *E. coli* show that native thioesterases led to premature termination of r-BOX intermediates, which also led to the formation of butanoic acid and difficulty in controlling r-BOX specificity[11].

We, therefore, optimized the extract background (i.e., used engineered *E. coli* source strains for extract preparation) to increase acetyl-CoA pools and decrease premature hydrolysis. Fortunately, previous work optimizing *E. coli* for r-BOX has been carried out to knock out side-reactions that (i) consume acetyl-CoA (i.e., Δ*pta*, Δ*adhE*), (ii) serve as alternative fermentation pathways during anaerobic growth (i.e., Δ*ldhA*, Δ*frdA*, Δ*sucD*, and Δ*poxB*), and (iii) lead to premature hydrolysis of the CoA-ester intermediates (i.e., Δ*yciA* Δ*ybgC* Δ*ydil* Δ*tesA* Δ*fadM* Δ*tesB*) or are involved in product degradation (Δ*fadE*). Thus, we made cell extracts from MB263 (DE3), MB263Δ*sucD* (DE3)[9], JC01 (DE3)[7], and JST07 (DE3)[11] that each carry a different combination of these knockouts (Source Data file). We first ensured that all extracts were proficient in CFPS with each making greater than 400 μg/mL of superfolder GFP (sfGFP)[22] (Supplementary Fig. S2A), and found that side-product formation of organic acids and alcohols in extracts sourced from these strains were significantly reduced compared to BL21*(DE3) (Fig. S2B). Using *E. coli* thioesterase TesA (Ec_tesA), we then tested the thioesterase activity in these extracts (Fig. 1C). Notably, JST07 containing six thioesterase knockouts (Δ*yciA* Δ*ybgC* Δ*ydil* Δ*tesA* Δ*fadM* Δ*tesB*) no longer showed butanoic acid formation and almost 10-fold increased hexanoic acid concentration $(6.6 \pm 0.4$ mM) compared to BL21*(DE3). This can be attributed to the reduction in premature termination of butyryl-CoA by the native thioesterases present in the other extracts. We then screened termination enzymes in JST07 and found Ec_tesA to be the best enzyme for hexanoic acid production (Supplementary Fig. 2C) In addition, JST07 is a Δ*pta* strain meaning that high-levels of acetate often present in crude lysate-based cell-free reactions no longer complicate mass-balance analysis of cell-free metabolic engineering assays. This was confirmed using $^{13}$C-labeled glucose to initiate reactions with and without the addition of Pta-enriched extract showing that $91 \pm 1\%$ of the carbon in hexanoic acid originates from glucose (Fig. 1D). We therefore chose to use the JST07 extracts for prototyping r-BOX using iPROBE.

We next automated the iPROBE framework to increase the throughput by downscaling the assay setup from a total reaction volume from 30 μL reactions to 4 μL reactions, allowing for assembly of r-BOX variants into 96-well plates using an acoustic liquid handling robot[23] (Supplementary Fig. 2E). Using the automated workflow, we optimized the major reaction components using the initial r-BOX enzyme set across 81 reaction conditions, which resulted in hexanoic acid yield of up to $8.2 \pm 2.0$ mM (Fig. 1E). We used these optimized conditions (5 mM CoA, 3 mM NAD$^+$, 0% fresh extract, and pH 7) as a starting point for the combinatorial screen of r-BOX enzyme homologs.

**High-throughput in vitro screen for hexanoic acid production.** We next used genome mining, phylogenetic analysis, and literature research combined with access to the sequences from the largest industrial collection of Acetone-Butanol-Ethanol (ABE) fermentation strains—the David Jones Collection—to identify a diverse set of 100 enzyme homologs for the four enzymes involved in initiation and elongation reactions of r-BOX (TL,

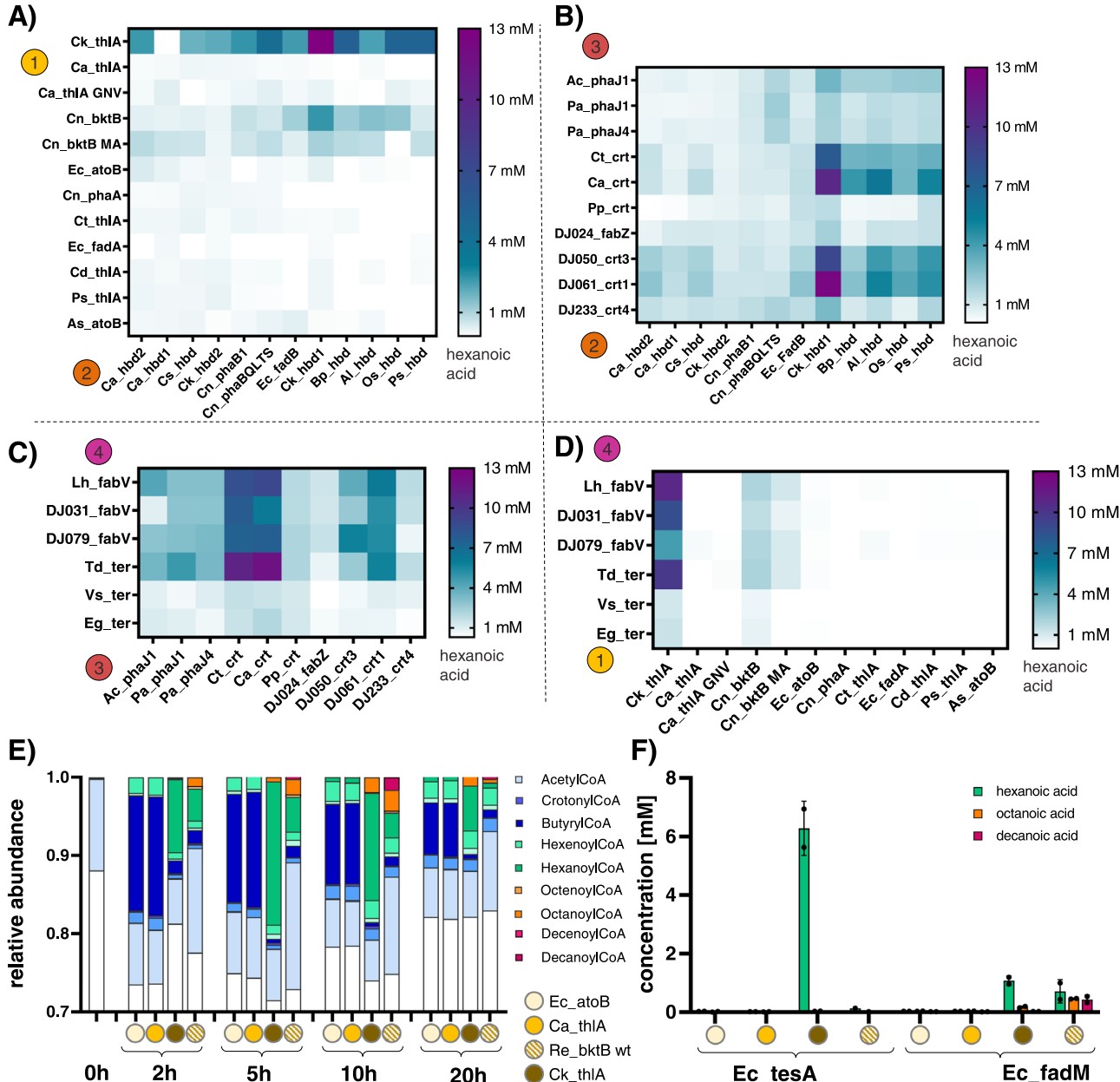

**Fig. 2 Combinatorial screen of r-BOX enzyme homologs.** All r-BOX enzymes were used at 0.3 μM with 0.15 μM of Ec_tesA and incubated at 30 °C for 24 h. $n = 3$, error bars = not shown (see Source Data file for complete data and enzyme abbreviations). The enzymes kept constant during the assay are Ck_thlA, Ck_hbd1, Ca_crt, Td_ter and Ec_tesA if they are not the ones investigated in the panel. **A** TL × HBD. **B** HBD × CRT. **C** CRT × TER. **D** TER × TL. Full results in Supplementary sheet 1. **E** Time-course SAMDI-CoA experiment showing dynamics of r-BOX assays over time. Maximum build-up of decanoyl-CoA occurs at 10 h using Re_bktB as the thiolase. All conditions were run and measured in quadruplicate and the average is displayed. **F** GC-MS characterization of the time course conditions with the addition of 0.15 μM of either Ec_tesA or Ec_fadM termination enzyme. Data represent $n = 3$ independent experiments, with the standard deviation and mean shown.

HBD, CRT, and TER) to be characterized (Supplementary Fig. 3A and Source Data file). These genes were codon optimized for *C. autoethanogenum*. Following gene synthesis, we carried out cell-free expression of the enzyme homologs and tested their soluble production (Supplementary Fig. 3B and Source Data file). Homologs expressing above 1 μM in CFPS were used in combinatorial screening of r-BOX pathway variants to assess activity. Select enzyme variants that worked well in other published systems but showed low solubility in CFPS were codon optimized for *E. coli* to increase expression and include these variants in the screen[24] (noted in Source Data file). We then stepped through a series of four screens varying two enzymatic steps at a time for a total of 392 unique enzyme combinations (Fig. 2). We first screened all TLs in combination with all HBDs and reasoned that due to the thermodynamic bottleneck of the thiolase reaction[5], efficient removal of the 3-ketoacyl-CoA intermediate was crucial for efficient reversal of β-oxidation (Fig. 2A). From these 144 different enzyme combinations, we identified the combination of Ck_thlA and Ck_hbd1 to produce more than 6 times more hexanoic acid than the optimized base case using Cn_bktB. We therefore used Ck_thlA as the new standard TL in subsequent screens (Fig. 2B–D). Overall, the combination of Ck_thlA, Ck_hbd1, Ca_crt, and Td_ter showed highest yields of hexanoic acid, and the choice of thiolase had the biggest influence on

hexanoic acid yields. We validated the top 18 conditions in hand-pipetted 30-μL reactions (Supplementary Fig. 2D, E) and further optimized the best combination by screening both the assay conditions (Supplementary Fig. 4A) and varied the concentrations of each enzyme in the pathway (Supplementary Fig. 4B). The best r-BOX iPROBE assay contained 0.6 μM of Ck_thlA, 0.6 μM of Ck_hbd1, 0.2 μM of Ca_crt, 0.6 μM of Td_ter, 0.1 μM of Ec_tesA, 4 mM of CoA, 3 mM of $NAD^+$ at pH 7 and improved in vitro hexanoic acid titers over 50-fold from our starting point to $38 \pm 5$ mM ($4.4 \pm 0.6$ $gL^{-1}$).

**Time-resolved detection of all cyclical intermediate of r-BOX.** After identifying a best set of enzymes for C6-product formation, we hypothesized that we could elucidate enzyme specificities and potential bottlenecks by analyzing CoA ester intermediates of r-BOX (4 per cycle) in a time-resolved manner using SAMDI-MS (Supplementary Fig. 5A)[25]. We first validated that purified CoA esters (butyryl-, hexanoyl-, octanoyl- and decanoyl-CoA) spiked into empty cell-free assays could be semi-quantitatively recovered (Supplementary Fig. 5B). However, the presence of the r-BOX termination enzyme (Ec_tesA) significantly reduced the amount of detectable CoA ester intermediates during pathway operation (Supplementary Fig. 5C). We, therefore, chose to further study the build-up of higher chain-length CoA ester intermediates using SAMDI-MS in the absence of a termination enzyme. We attempted to elucidate chain-length specificity of the optimized pathway by measuring the build-up of CoA ester intermediates over time (Fig. 2E). We found that thiolase selection had a significant effect on the build-up of CoA ester intermediates, with Ck_thlA displaying high specificity for hexanoyl-CoA. We were able to further modulate specificity by co-expressing combinations of thiolases (i.e., co-addition in vitro) (Supplementary Fig. 5D, E), which would have been difficult to evaluate in vivo, providing even more evidence that thiolase selection is essential for controlling both initiation and elongation. While we observed C6-chain length specificity of Ck_thlA in our iPROBE screen of fatty acids (Fig. 2A), we did not detect longer chain fatty acids which we would have expected based on our measurement of CoA ester intermediates (Fig. 2E). We surmised that this is due to the substrate specificity of Ec_tesA, the thioesterase used for cycle termination. We, therefore, tested the conditions for longer chain termination with Ec_fadM, a thioesterase with longer-chain substrate specificity, and were indeed able to detect the formation of both octanoic and decanoic acid by GC-MS (Fig. 2F). Thus, coupling the selection of thiolase and termination enzymes is important for controlling the product spectrum of r-BOX.

**Initiation and termination enzymes lead to C4- and C6-specificity.** With a robust core set of r-BOX enzymes capable of producing high amounts of the two-cycle product (hexanoic acid) in hand, we decided to further explore the termination enzyme in controlling product specificity[11,19]—some combinations might prefer termination after one cycle of r-BOX leading to C4-products versus two cycles, C6-products, and higher. We, therefore, screened the six best performing thiolases against two thioesterases and four phosphate butyryltransferases (Ptb) in vitro and quantified both butanoic and hexanoic acid production (Fig. 3A). The Ptb enzymes in combination with a butyrate kinase (Buk) form a clostridia-native ATP-conserving termination system, which could lead to more efficient production in non-model organisms like *C. autoethanogenum*. The Ptb-Buk systems performed well for their native substrate and the best combination of Ca_thlA_GNV and Cb_ptb produced $30.0 \pm 0.2$ mM butanoic acid while producing only $0.21 \pm 0.03$ mM hexanoic acid. Ck_thlA was important for high

hexanoic acid production and its combination with Ec_tesA produced $20.4 \pm 0.6$ mM hexanoic acid with no detectable butanoic or octanoic acid. Using the same approach, we then screened thioesterases against four acyl-CoA reductases (ACR) in combination with the alcohol dehydrogenase Aa_adh[26] and two bifunctional alcohol/aldehyde dehydrogenases (AdhE) a while keeping the HBD, CRT, TER core constant to find r-BOX variants able to produce butanol (one cycle) and hexanol (two cycles) at high specificity and titers (Fig. 3B). The best butanol variant containing Ec_atoB and the bifunctional Ca_adhE produced $20.8 \pm 0.3$ mM butanol while only producing traces of hexanol (Source data file). Ck_thlA was again the main determinant for C6 product formation and the combination with the ACR Maqu2507 produced $4.8 \pm 0.4$ mM hexanol with no detectable levels of butanol or octanol. With iPROBE, we were able to determine enzyme sets for C4- and C6-specific production of acids and alcohols.

**iPROBE r-BOX variants yield *E. coli* strains with high selectivity.** We next explored the ability of our cell-free-selected r-BOX pathway designs to translate to high- and specific-producers of C4- and C6-acids and alcohols in heterotrophic (*E. coli*) and autotrophic (*C. autoethanogenum*) organisms, starting with *E. coli*. To do this, we selected 40 unique pathway combinations from our screens and tested them in *E. coli* JST07 for butanoic and hexanoic acid (Fig. 4A), as well as butanol and hexanol (Fig. 4B), production. The best hexanoic acid-producing variant using Ck_thlA and Ec_tesA produced $26.3 \pm 0.3$ mM ($3.06 \pm 0.07 gL^{-1}$), which to our knowledge represents the highest demonstrated titers achieved in *E. coli* to date. Overall, the in vitro product spectrum of the tested combinations correlated well ($r = 0.92$, Source Data file) with the spectrum observed in *E. coli* (Fig. 4C). However, the alcohol-producing strains showed larger differences. Cb_ald generally did not perform well in vitro most likely due to low solubility, and Ca_adhE did not perform well in *E. coli* which was expected from previous in vitro studies[11]. The combination of Ck_thlA and Cb_ald surprisingly produced the highest butanol titers ($11 \pm 3$ mM or $0.8 \pm 0.2$ $gL^{-1}$) even though our in vitro data suggested Ck_thlA would produce high hexanol titers (Fig. 4D). The best hexanol-producing condition consisted of Ec_fadB, Td_ter in combination with Ck_thlA and Ec_mphF. With this strain, we were further able to increase hexanol production two-fold using a dodecane overlay to remove hexanol during growth to mitigate any toxic effects[27] (Fig. 4E). The iPROBE-optimized enzyme set was both the highest and most specific C6-alcohol producer, with three-fold lower amounts of octanol produced than our initial enzyme set.

**r-BOX in *C. autoethanogenum* leads to hexanol production.** We next tested whether our iPROBE-selected r-BOX pathway variants also improved specificity for C4- and C6-alcohols in autotrophic, anaerobic *C. autoethanogenum*. In previous work[24], we have established a modular vector system that allows us to build up *Clostridium* in vivo expression constructs directly from cell-free vector by combining with different parts such as promoters of different strength. Note that the native aldehyde ferredoxin oxidoreductase (Ca_aor)[28] reduces the produced acids to the corresponding alcohols. Thioesterase termination enzymes were therefore included in screening for strains that produce C4- and C6-alcohols. Knowing the importance of the thiolase and the *C. autoethanogenum* native more efficient bcd-etf system that replaces Td_ter in r-BOX[12] (previously shown to increase titers for butanol production), we first tested impact of expression of the iPROBE-selected thiolase (Ck_thlA) and the Bcd-Etf system in *C. autoethanogenum*[12] while keeping the rest of the core enzymes (i.e., Ck_hbd1, Ca_crt, Ec_tesA) constant. For this we

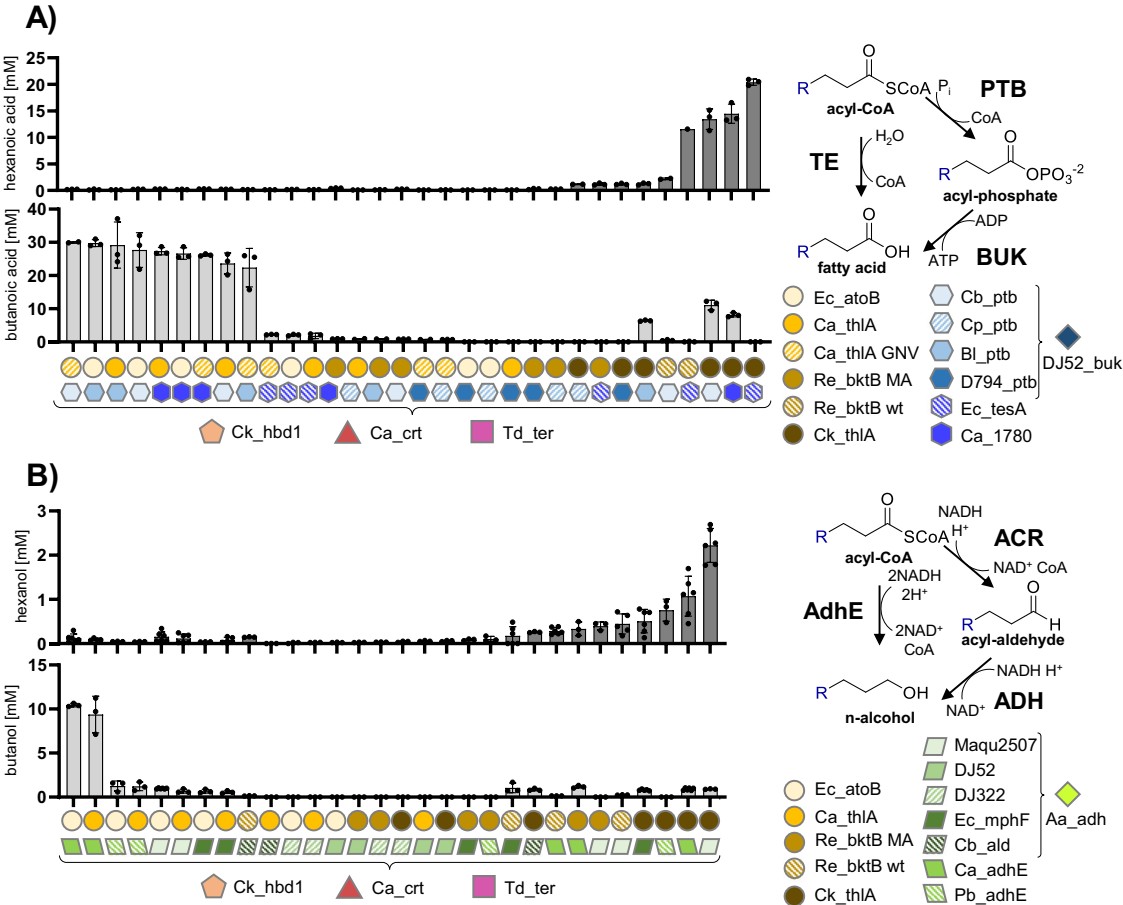

**Fig. 3 In vitro screening of thiolase and termination enzymes to find specific variants for C4 and C6 products using the r-BOX core enzymes found in Fig. 2. A** Thioesterase and ptb-buk termination screen for the production of short-chain fatty acids. All assays contained each enzyme at 0.3 µM concentration. DJ52_buk was added to the enzymes containing ptb's. **B** ACR-ADH and bifunctional AdhE screen for the production of butanol and hexanol. All assays contained thiolase and termination enzymes at 0.3 µM concentration and the three core enzymes Ck_hbd1, Ca_crt, and Td_ter at 0.15 µM due to volume limitations. Assays with an ACR were complemented with 0.15 µM of Aa_adh. All measured data points and their mean are shown with n greater than or equal to 3, and error bars represent standard deviation. Detailed results and full information on enzyme abbreviations in Source Data file.

used the promoters already onboarded to our modular vector system, a *C. autoethanogenum* ferredoxin promoter (Pfer or Pfdx), a *C. autoethanogenum* phosphotransacetylase promoter (Ppta), a *C. autoethanogenum* pyruvate:ferredoxin oxidoreductase promoter (Ppfor) and *C. autoethanogenum* Wood-Ljungdahl cluster operon (Pwl or PacaA)[24]. The strain containing Ck_thlA under control of Pwl produced the highest titers of butanol and hexanol suggesting a fully active r-BOX cycle (Fig. 5A). We used this architecture and a previously reported 'cell-free to cell' vector system[24] to generate 14 strains with iPROBE-selected r-BOX combinations for either high butanol or hexanol production. The strains containing the Ca_thlA thiolase showed high production of butanol ($1.71 \pm 0.02$ mM) as predicted from its C4 specificity both in vitro and in *E. coli* (Fig. 5B), more than a 7-fold improvement over the best producing *C. autoethanogenum* strain observed in a previous screening effort[12]. However, the Ptb-Buk system underperformed in *C. autoethanogenum*. To address this, we replaced the Pfer promoter with the PwlI promoter, as this previously helped with the Ck_thlA thiolase, which led to a > 10-fold increase in butanol production (Fig. 5C). The strain containing Ck_thlA and the ACR Maqu2507 produced the highest titers of hexanol with $0.54 \pm 0.09$ mM (Fig. 5B). This is, to our knowledge, the first report of hexanol production in *C. autoethanogenum* and in vitro prototyping was crucial in finding a functional thiolase (Fig. 5D). While the in vitro-in vivo

correlation could be improved ($r = 0.46$, Source Data file), in vitro prototyping coupled with strain and promoter optimization has potential to reduce in vivo testing iterations.

The best-performing iPROBE-selected strain for hexanol production was chosen for process scale-up from 0.01 L bottle fermentations to 1.5 L continuous fermentations using syngas ($CO/H_2/CO_2$) as the sole carbon and energy source. Over a 2-week fermentation, we monitored hexanol and biomass in a control strain and our iPROBE-selected strain with and without optimized fermentation conditions (Fig. 5E). In fermentations, we observed hexanol titers of ~200 mg L$^{-1}$ (2.0 mM) at a rate of ~125 mg L$^{-1}$ d$^{-1}$ in a continuous system. *C. carboxidivorans* a related acetogenic clostridia has recently been shown to natively produce butanol, butanoic acid, hexanol, hexanoic acid and the product spectrum in that organism is heavily growth condition dependent[29,30]. While our study explored numerous enzyme sequences, further exploration of enzymes from *C. carboxidivorans* P7 may provide further benefits. This suggests that yields and specificity of our generated *C. autoethanogenum* strains can be further improved through growth condition optimizations.

**Discussion**

Here, we use high-throughput in vitro enzyme prototyping (iPROBE) to optimize r-BOX product specificity and compare

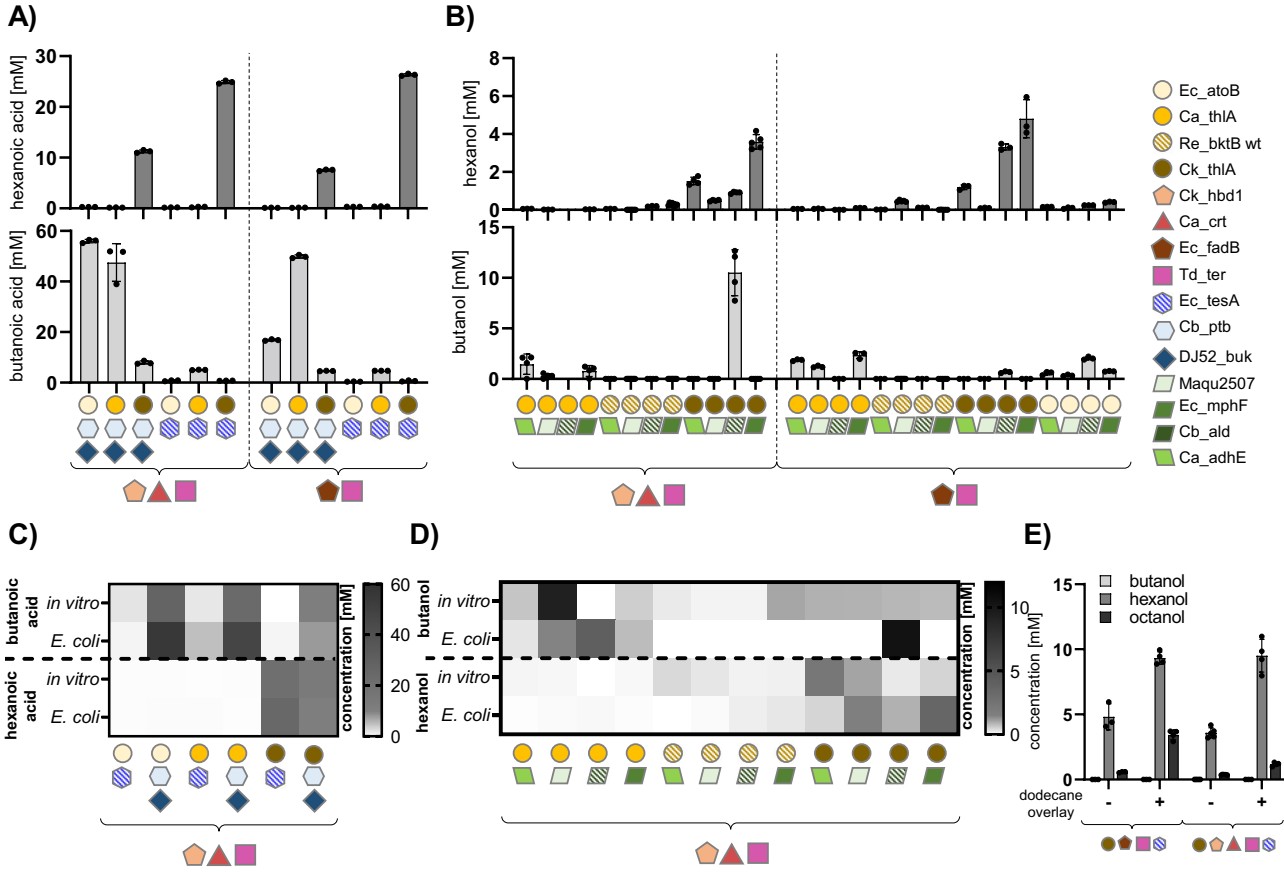

**Fig. 4 Testing r-BOX variants in *E. coli*. A** Quantification of butanoic and hexanoic acid production in *E. coli* strains containing newly identified core enzymes (left) or previously characterized enzyme combination (right). **B** Quantification of butanol and hexanol production in *E. coli* strains containing newly identified core enzymes (left) or previously characterized enzyme combination (right), various thiolases and a range of ACR-ADH or AdhE termination enzymes. **C** Direct comparison of in vitro (see Fig. 3A) and *E. coli* butanoic and hexanoic acid production (see **A**). **D** Direct comparison of in vitro (see Fig. 3B) and *E. coli* butanol and butanoic acid production (see **B**). **E** Strains containing Ck_thlA and Ec_tesA as well as the previously characterized old core enzymes (left) or new core enzymes (right) with or without a dodecane organic overlay. Assays were run in at least triplicates (*n* > 3) and all data points and their mean are shown. Error bars represent standard deviation. Detailed results and full information on enzyme abbreviations in Source Data file.

r-BOX pathway performance across three different platforms—a cell-free system, *E. coli* as a heterotrophic model organism, and autotrophic *C. autoethanogenum* capable of using syngas as the sole carbon and energy source. Using the in vitro iPROBE system as a guide, we were able to significantly improve r-BOX production, achieving the highest reported titers and/or selectivities towards C6 r-BOX products in *E. coli* and demonstrating r-BOX for the first time in *C. autoethanogenum* from syngas[11,31,32].

Our approach has several key features. First, we automated iPROBE using an acoustic liquid handling robot to increase the throughput power of this approach. We used this automated approach to show that extracts from knockout strains are critical to increase starting substrate pools (i.e., acetyl-CoA), reduce side product formation, control carbon flux (i.e., enable theoretical yield calculations), and remove competing background reactivities. The developed JST07 extract can be adapted for use for iPROBE prototyping of a wide range of biosynthetic pathways that use CoA ester intermediates and start with acetyl-CoA or pyruvate (e.g., isoprenoids, fatty acids, cannabinoids, polyketides). Additionally, it has the potential to enable faster optimization of recently established CoA ester-dependent new-to-nature/synthetic pathways like CETCH[33] and FORCE[34].

Second, by using this approach, we were able to tailor r-BOX to produce a single product at high selectivity. As is generally the case for iterative pathways, r-BOX typically generates a product mixture[11,31,32]. Here we identified designs that specifically favor either C4 or C6 products with the termination step playing a key role. Another key step in the r-BOX pathway is the initial priming step, which is a thermodynamically challenging carbon-carbon bond forming thiolase reaction. Efficient removal of the thiolase product by HBD additionally helps to overcome this bottleneck.

Third, we compared designs across three platforms. Despite the distinct differences between the three systems (e.g., no cofactor, salt, inorganic phosphate homeostasis in vitro, low activity of the bifunctional alcohol-aldehyde dehydrogenases AdhE in *E. coli*, utilization of the oxygen-sensitive ferredoxin dependent Bcd-Etf and Aor enzymes in *C. autoethanogenum*), we show that iPROBE can accelerate cellular design especially when prototyping complex pathways for non-model organisms. The r-BOX variants identified as good candidates for butanol, butanoic acid, hexanol, and hexanoic acid synthesis translated well to the two in vivo systems and enabled the *E. coli* strains with the highest and most specific demonstrated titers of hexanol and hexanoic acid products achieved in *E. coli* to date as well as the synthesis of hexanol in the acetogenic *C. autoethanogenum*. This sets a good starting point for applying iPROBE to other metabolic pathways, while keeping in mind that inherent differences between in vitro and in vivo systems may not always allow direct correlation across these systems. For example, anaerobicity of *C. autoethanogenum* and dependence on oxygen-sensitive enzymes (that cannot be prototyped in in vitro systems so far).

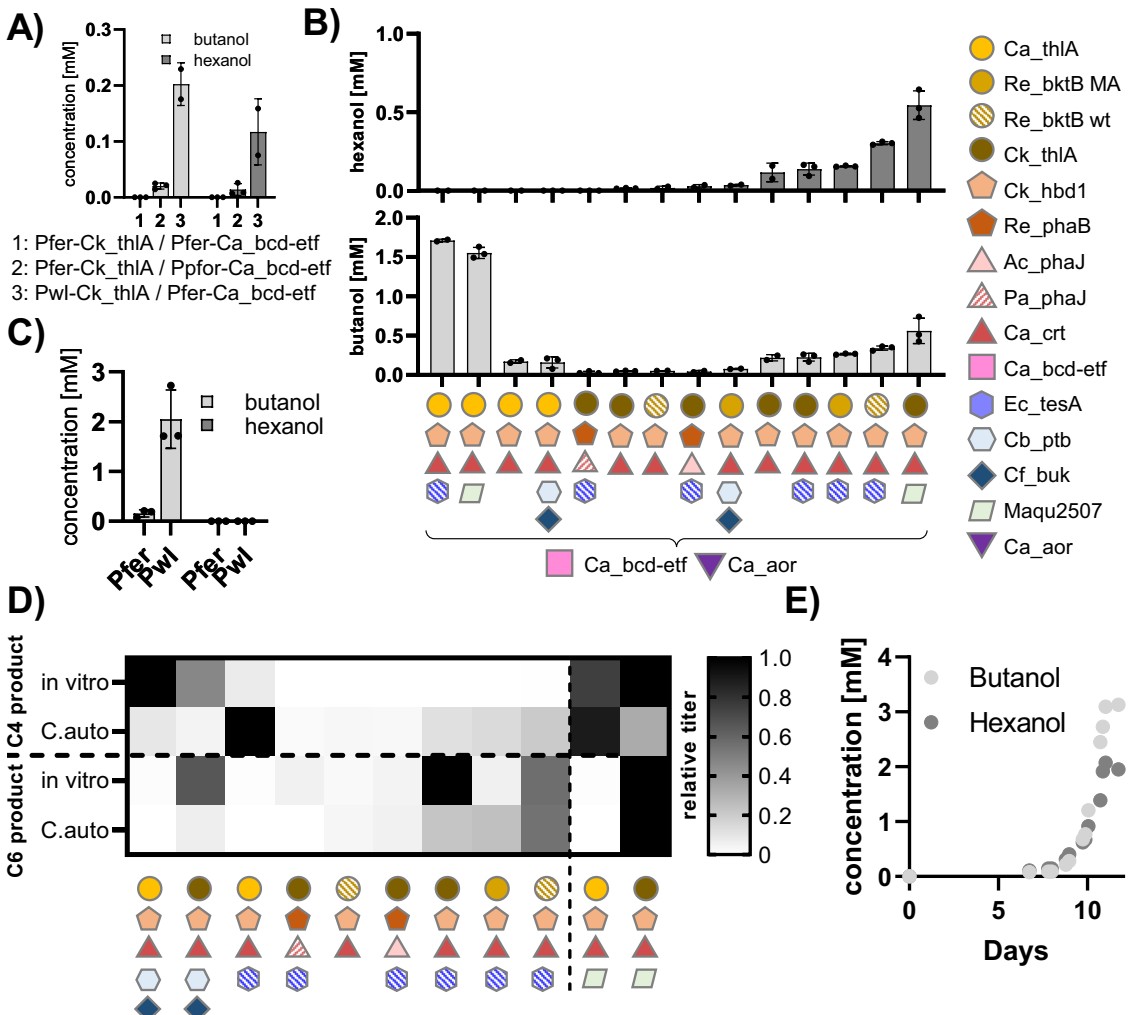

**Fig. 5 Implementation of r-BOX into the non-model organism *C. autoethanogenum*. A** Initial promoter optimization to define architecture for the screen. **B** Testing 14 different r-BOX variants in *C. autoethanogenum* for butanol and hexanol production. For detailed strain architecture see attached sheet Figure5_Cauto. **C** Promoter optimization for *C. autoethanogenum* strain containing Ca_thlA, Ck_hbd1, Ca_crt, Ca_bcd-etf, Cb_ptb, Cf_buk. **D** Normalized titers for C4 and C6 products in vitro and in *C. autoethanogenum*. In vitro data was normalized to the highest producing condition observed in Fig. 2 for either acid (in case of thoesterase or ptb-buk termination) or alcohol (in case of ACR-ADH termination). *C. autoethanogenum* titers were normalized to the highest alcohol producing strain for either butanol or hexanol. Data represent at least $n = 3$ independent experiments, with the standard deviation and mean shown. Detailed results and full information on enzyme abbreviations in Source Data file. **E** Continuous 1.5 L fermentation of *C. autoethanogenum* strain containing Ck_thlA, Ck_hbd1, Ca_crt, Ca_bcd-etf, Ec_tesA.

Looking forward, our strategy of in vitro prototyping, optimization in model organisms, and implementation into production hosts as well as the observed correlations between the systems can be widely utilized to reduce the development time of new industrial strains and enable optimization a wide range of r-BOX derived products.

## Methods

**Materials**. DNA for all enzymes used in this study was ordered from Twist Bioscience (CA, USA) or provided by JGI in the vectors pJL1 (Addgene #69496), pD1, pD3, or pD5[24]. *C. autoethanogenum* codon optimization was generated using LanzaTech's in-house codon optimization software, *E. coli* optimization was done using the IDT codon optimizer GENEious. All consumables were purchased from Sigma-Aldrich unless stated otherwise. Standard microtiter plates (96- and 384-well) were purchased from Corning. WebSeal 96 glass-coated well plates were purchased from ThermoFisher. The *E. coli* strain BL21(DE3) was obtained from NEB, all other strains were provided by the Gonzales lab.

**Cell-free extract preparation**. BL21 (DE3) was prepared according to previously described methods[35]. All other extracts were prepared with a slightly adjusted protocol after optimizing yields of superfolder GFP (sfGFP) produced in CFPS[22].

1 L of 2x YTPG medium (pH set to 7.2) was inoculated with an overnight culture of *E. coli* JST07 at an $OD_{600}$ of 0.1 and then grown at 37 °C to an $OD_{600}$ of 1.8. All subsequent steps were performed on ice. Cells were harvested by centrifugation at $8000 \times g$ for 10 min, washed three times with 30 mL S30 buffer (for cell-free protein synthesis), and then resuspended in 1 mL S30 buffer per gram of cell pellet. Cells were lysed using a high-pressure liquid homogenizer, 2.85 mM DTT was added to the lysate, followed by a centrifugation of $12,000 \times g$ for 10 min. Resulting supernatant was then incubated at 37 °C followed by a second centrifugation step at $12,000 \times g$ for 10 min. Resulting supernatant was aliquoted, flash frozen in liquid nitrogen, and stored at −80 °C.

**Cell-Free Protein Synthesis Reactions**. All CFPS reactions used a modified PANOx-SP formula[20]. Reactions included 6 mM magnesium glutamate, 10 mM ammonium glutamate, 134 mM potassium glutamate, 1.2 mM ATP, 0.85 mM GTP, UTP and CTP each, 0.034 mg mL$^{-1}$ folinic acid, 0.171 mg mL$^{-1}$ *E. coli* tRNA mixture, 33.3 mM PEP, 2 mM of each amino acid, 0.33 mM NAD$^+$, 0.27 mM coenzyme A, 1 mM putrescine, 1.5 mM spermidine, 57 mM HEPES pH 7.2, 0.1 mg mL$^{-1}$ T7 RNA polymerase, 13.3 ng mL$^{-1}$ and 26.6 v/v% JST07 extract. Reactions were incubated for 20 h at 37 °C. CFPS yields of all r-BOX proteins were determined by $^{14}$C leucine incorporation using previously described protocols[12].

**Cell-free metabolic engineering assays**. Assays were assembled using an Echo 550 liquid handling robot (Labcyte Inc. USA) into 96- or 384-well plates. The

reaction volume was 4 L with 2 L consisting of CFPS reaction (CFPS reaction mix without plasmid was added to normalize volume of all the conditions), 6 mM magnesium acetate, 10 mM ammonium acetate, 134 mM potassium acetate, 120 mM glucose, 10 mM BisTris buffer pH 7, 3 mM $NAD^+$, 0.5 mM kanamycin, 4 mM coenzyme A, and 0.1 U/mL catalase from *Aspergillus niger* (Sigma Aldrich), unless stated otherwise.

**GC-MS analysis**. For GC-MS analysis, reactions were run in glass-coated WebSeal 96-well plates (ThermoFisher), quenched with 10 mL quenching mix (15% NaCl, 12.5% sulfuric acid, and 2 mM nonanoic acid [internal standard]), and extracted with 200 mL hexane. 30 mL of the hexane fraction was derivatized with 5 mL pyridine and 5 mL *N,O*-Bis(trimethylsilyl)trifluoroacetamide at 70 °C in glass vials. Samples were then analyzed on an Agilent HP-5MS (30 m length × 0.25 mm i.d. × 0.25 μm film) column with helium carrier gas at constant flow of 1 mL min$^{-1}$. The inlet temperature was 250 °C and column temperature started at 50 °C, held for 2 min, then increased at 60 °C min$^{-1}$ to 190 °C, then increased at 120 °C min$^{-1}$ to 270 °C, and was held for 5 min. Injection volume was 1 μL with a split ratio of 4:1. Concentrations were determined by comparing it to a freshly prepared standard curve of fatty acids extracted in parallel with the samples from iPROBE reactions containing no plasmid.

**HPLC analysis**. HPLC was used to analyze glucose, pyruvate, succinate, lactate, butanoic acid, and butanol using an Agilent 1260 series HPLC via a refractive index detector. Samples were quenched with 12 μL of 50% sulfuric acid, centrifuged and the supernatant transferred to new 96 well plates. 10 μL of sample was separated on an Aminex HPX-87H column (Bio-Rad, Hercules, CA) at 55 °C using 5 mM $H_2SO_4$ as a mobile phase with a flow rate of 0.6 mLmin$^{-1}$. Metabolite concentrations were determined by comparison to a calibration curve of standards treated the same way using Agilent OpenLab software.

**SAMDI-CoA analysis**. Sample preparation and analysis were done according to the previously reported method[25,36]. In short, assays run in 96 or 384 well plates were quenched with 4 μL 5% formic acid, diluted with 12 μL of water, flash frozen, and lyophilized overnight. Samples were then derivatized with capture peptide according to the reported methods and analyzed using an AB Sciex 5800 MALDI TOF/TOF instrument in positive reflector mode.

**in vivo production of acids and alcohols in *E. coli***. JSTO7(DE3), a previously engineered derivative of *Escherichia coli* K12 strain MG1655[11], was used as the host for all *E. coli*-based product synthesis. Plasmid-based gene overexpression of r-BOX enzymes was achieved by cloning the desired gene(s) into either pETDuet-1 or pCDFDuet-1 (Novagen, Darmstadt, Germany) digested with appropriate restriction enzymes utilizing In-Fusion PCR cloning technology (Clontech Laboratories, Inc., Mountain View, CA). Cloning inserts were created via PCR of ORFs of interest from their respective genomic (all *E. coli* genes) or codon-optimized (all other genes) DNA with Phusion polymerase (Thermo Scientific, Waltham, MA). The resulting In-Fusion products were used to transform E. coli Stellar cells (Clontech Laboratories, Inc., Mountain View, CA) and PCR identified clones were confirmed by DNA sequencing.

The minimal medium designed by Neidhardt et al.[37], with 125 mM MOPS and Na$_2$HPO$_4$ in place of K$_2$HPO$_4$, supplemented with 20 g/L glycerol, 10 g/L tryptone, 5 g/L yeast extract, 100 μM FeSO$_4$, 1.48 mM Na$_2$HPO$_4$, 5 mM (NH$_4$)$_2$SO$_4$, and 30 mM NH$_4$Cl was used for all fermentations. Antibiotics (50 μg/mL carbenicillin, 50 μg/mL spectinomycin) and inducer (20 μM isopropyl β-D-1-thiogalactopyranoside, IPTG) were included for all experiments. All chemicals were obtained from Fisher Scientific Co. (Pittsburg, PA) and Sigma-Aldrich Co. (St. Louis, MO). Fermentations were conducted in 25 mL Pyrex Erlenmeyer flasks (Corning Inc., Corning, NY) filled with varying volumes (10 mL for acid production and 20 mL for alcohol production) of the above culture medium and sealed with foam plugs filling the necks. A single colony of the desired strain was cultivated overnight (14–16 h) in LB medium with appropriate antibiotics and used as the inoculum (1%). After inoculation, flasks were incubated at 37 °C and 200 rpm in a New Brunswick I24 Benchtop Incubator Shaker (Eppendorf North America, Enfield, CT) until an optical density of ~0.3–0.5 was reached, at which point IPTG was added. For hexanol production fermentations including an organic overlay, 5 mL of dodecane was added to 15 mL of the culture medium at the time of induction. Flasks were then incubated under the same conditions for 48 h post-induction.

Concentrations of glycerol, ethanol, and organic acids (<C5) were determined via HPLC using a Shimadzu Prominence SIL 20 system (Shimadzu Scientific Instruments, Inc., Columbia, MD) equipped with a refractive index detector and a HPX-87H organic acid column (Bio-Rad, Hercules, CA) with the following operating conditions: 0.3 mL/min flow rate, 30 mM H$_2$SO$_4$ mobile phase, column temperature 42 °C. Quantification and identification of additional compounds were carried out using an Agilent Intuvo 9000 gas chromatography system equipped with a 5977B inert plus mass selective detector turbo EI bundle (for identification), a flame ionization detector (for quantification), and an Agilent HP-5ms capillary column (0.25 mm internal diameter, 0.25 μm film thickness, 30 m length) as previously described[38].

**In vivo production in *Clostridium autoauthanogenum***. *C. autoethanogenum* codon optimized genes were cloned into pMTL8000 Clostridial expression vectors via Golden Gate assembly as previously described[24]. Resulting plasmids were sequence confirmed and transformed in *C. autoethanogenum* using protocols used previously[39]. Resulting strains were grown and tested in small-scale bottles per protocols described earlier using a synthetic gas blend consisting of 50% CO, 10% H2, 40% CO2 (Airgas, Radnor, PA)[39]. Continuous fermentations were carried out in 1.5 L continuous stirred tank reactors (CSTRs) with constant gas flow as described previously[28].

**Reporting summary**. Further information on research design is available in the Nature Research Reporting Summary linked to this article.

## Data availability

All data presented in this manuscript are available as Source Data file. Materials are available upon reasonable request and under MTA, but strains may require a license.

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

## Acknowledgements

This work was supported by the Office of Energy Efficiency and Renewable Energy (EERE) under Contract No. DE-EE0008343 and the U.S. Department of Energy (DOE) Biological and Environmental Research Division (BER), Genomic Science Program (GSP) for funding of this project under Contract No. DE-SC0018249. DNA synthesis for the gene libraries was supported by the Joint Genome Institute Community Science Program under award no. CSP-503280; https://doi.org/10.46936/10.25585/60001121 (M.C.J. and M.K.). The work conducted by the U.S. Department of Energy Joint Genome Institute (https://ror.org/04xm1d337), a DOE Office of Science User Facility, is supported by the Office of Science of the U.S. Department of Energy operated under contract no. DE-AC02-05CH11231. M.C.J. acknowledges the David and Lucile Packard Foundation. We also thank the following investors in LanzaTech's technology: BASF, CICC Growth Capital Fund I, CITIC Capital, Indian Oil Company, K1W1, Khosla Ventures, the Malaysian Life Sciences, Capital Fund, L. P., Mitsui, the New Zealand Superannuation Fund, Novo Holdings A/S, Petronas Technology Ventures, Primetals, Qiming Venture Partners, Softbank China, and Suncor. B.V. acknowledges support from the Swiss National Science Foundation via a SNSF Early Postdoc.Mobility fellowship (P2SKP3_184036). B.R.K. acknowledges support from the Ryan Fellowship, the International Institute for Nanotechnology at Northwestern University. L.S. was supported by the Zeno Karl Schindler Foundation.

## Author contributions

B.V., L.S., S.G., K.T., J.M.C., S.D.B, S.D.S., M.M., A.S.K., R.G., M.C.J., and M.K. designed the study; B.V. and L.S. performed all cell-free experiments; S.G., A.G., L.T., and H.Z. performed clostridia experiments; K.T. and J.M.C. performed *E. coli* experiments; E.H.M. and B.R.K. performed SAMDI-CoA experiments; B.V., L.S., and A.S.K. performed data analysis and visualization. B.V., L.S., S.G., J.M.C., M.M., A.S.K., R.G., M.K., and M.C.J. wrote the manuscript; All authors edited and reviewed the manuscript.

## Competing interests

M.K., S.G., J.M.C., S.D.B., S.D.S., L.T., H.Z., and A.G. are current employees of Lanza-Tech, a for-profit company pursuing commercialization of the *C. autoethanogenum* gas fermentation process discussed here. M.C.J. and R.G. consult for and have joint funding with LanzaTech. R.G. is the sole proprietor of RBN Biotech LLC, which holds rights to several r-BOX patents. All other authors declare no competing interests.
