## [Peer Review File · Nature Communications]

Reviewers' Comments:

Reviewer #1:

Remarks to the Author:

Vögeli et al. reported the implementation of r-BOX for synthesis of biochemicals, particularly, C4-C6 acids and alcohols. High-throughput in vitro prototyping has been carried out based on their previously established iPROBE procedure, and key enzyme sets were identified to enhance product selectivities. Further, the identified pathways were implemented in heterotrophic *E. coli* and autotrophic *C. autoethanogenum*. The identified pathway likely works better in *E. coli* than in *C. autoethanogenum*. This study was built upon their previous work on biochemical production through r-BOX and also the cell-free prototyping procedure, and has presented a huge amount of work.

Some specific comments as bellow:

1. In Fig. 1 A), at the one O'clock position, "acetyl-CoA" can be used instead of "acyl-CoA", to represent the "initiation elongation".
2. In Fig. 1, and all other figures, as well as in the main text of the manuscript: the authors should define the abbreviated enzyme names (or have a detailed definition in the supplementary materials and then refer to it in the main text and figures). Otherwise, it is very hard for readers to understand and follow. Along this line, Line 162: define "sfGFP".
3. Fig. 1D: 13C is in black color; 12C is in grey color. This should be described in the Figure caption.
4. Line 220 and Fig. 2: The names of the relevant enzymes in the text should be consistent with the one in the figure: Ck_thIA in Line 220 (while in the figure it is Ck_thIA1) // Ck2_hbd in Line 220 (while it is Ck_hbd1 in Figure 2).
5. Line 297 (Caption of Figure 3): DJ52_buk: same as comment #2, please define this enzyme name..
6. Figure 3 and also other figures: the colors used to represent different enzymes are very close and similar. The authors might want to revise the color code for a better differentiation of various enzymes..
7. Line 284: delete "a".
8. Line 305: For the *E. coli* work, what promoters/RBS are used.. Why did not further optimize the promoter/RBS usage to possibly further improve the product production levels?
9. Along this line, in Dellomonaco et al. (Nature, 2011), the engineered *E. coli* strain "synthesized 2.2 g/L of n-butanol in 24 h at a yield of 0.28 g n-butanol per g total glucose consumed. When grown in a bioreactor using a higher concentration of glucose, this strain produced n-butanol at high titre of 14 g/L, yield (0.33 g n-butanol per g total glucose consumed) and rate (2 g n-butanol per g cell dry weight per h)". While the butanol production level in *E. coli* in this study was much lower than this previously reported level even after extensive iPROBE-selection and optimization. How would the authors comment on this?
10. Line 339: For the work in *C. autoethanogenum*, why the acid production in *C. autoethanogenum* was not attempted here, or it has been attempted but the results were not good (and thus not presented)?.
11. Line 348: Pfer, Pwl: please define.
12. Fig. 5B: "Cf_buk" in the symbol (color code), but was actually not used in the figure.
13. Line 424: Aor: please define.
14. Line 473: "the acetone and isopropanol gas fermentation process discussed here" ??

Reviewer #2:

Remarks to the Author:

In this study, an iProbe rapid screening system was established in vitro, and verified in *Escherichia coli* and *C. autoethanogenum*, achieving very good results, which is very enlighten for the previous research. But a few additions are needed.

1. According to the description of the article, the reviewer understands that iProbe system is an application mode of cell-free protein expression system. May I ask if this understanding is correct? It is hoped that the author can draw a schematic diagram and briefly describe the system in the article.

2.Line 142. What's the PANox-SP CFPS system?Hope the author can make brief explanation to this abbreviation in the article.

3.Line 154. Acetyl-coa synthesis pathway is regulated by strict feedback inhibition. Can the inactivation of acetyl-CoA consumption pathway increase acetyl-CoA concentration?Is there a documentary basis?

4.Why do you choose *C. Autoethanogenum* as one of the validation platforms?

Point-by-point response to reviewers – Vogeli et al. 2022

Responses are highlighted in blue

Reviewer #1 (Remarks to the Author):

Vögeli et al. reported the implementation of r-BOX for synthesis of biochemicals, particularly, C4-C6 acids and alcohols. High-throughput in vitro prototyping has been carried out based on their previously established iPROBE procedure, and key enzyme sets were identified to enhance product selectivities. Further, the identified pathways were implemented in heterotrophic *E. coli* and autotrophic *C. autoethanogenum*. The identified pathway likely works better in *E. coli* than in *C. autoethanogenum*. This study was built upon their previous work on biochemical production through r-BOX and also the cell-free prototyping procedure, and has presented a huge amount of work.

We appreciate the kind words and detailed input of the reviewer and agree that rBOX currently seems to work better in *E. coli* than in *C. autoethanogenum*. This is in our opinion mostly due to the fact that a lot of engineering efforts have already gone into optimizing *E. coli* strains for rBOX (e.g. Dellomonaco et al., Kim et al., Cheong et al.). Our manuscript is to our knowledge the first report of C4 and C6 alcohol production in *C. autoethanogenum*. We are optimistic that the titers and production rates can be improved using similar strain engineering efforts as recently published for acetone (Liew et al. Nature Biotechnology, 2022) now that a starting point has been established. We are excited about this prospect and grateful to the reviewer for recognizing the huge amount of work that went into this story.

Some specific comments as bellow:

1. In Fig. 1 A), at the one O'clock position, "acetyl-CoA" can be used instead of "acyl-CoA", to represent the "initiation elongation".

That is more precise. Thank you for pointing this out. We have made the appropriate change.

2. In Fig. 1, and all other figures, as well as in the main text of the manuscript: the authors should define the abbreviated enzyme names (or have a detailed definition in the supplementary materials and then refer to it in the main text and figures). Otherwise, it is very hard for readers to understand and follow. Along this line, Line 162: define "sfGFP".

All abbreviated enzyme names are described in detail in 'Supplementary sheet 1 - Enzyme list' including their full DNA and amino acid sequence. We now refer to this list in both the text (Line 140) and in the Figure descriptions. sfGFP is now defined and a citation for the original publication added.

3. Fig. 1D: 13C is in black color; 12C is in grey color. This should be described in the Figure caption.

This is now described in the Figure caption.

4. Line 220 and Fig. 2: The names of the relevant enzymes in the text should be consistent with the one in the figure: Ck_thIA in Line 220 (while in the figure it is Ck_thIA1) // Ck2_hbd in Line 220 (while it is Ck_hbd1 in Figure 2).

Thank you for pointing out those inconsistencies. The Text, Figures, and Supplementary table have been updated to always refer to Ck_thIA (for Ck_thIA1) and to Ck_hbd1 (for Ck_hbd and Ck2_hbd).

5. Line 297 (Caption of Figure 3): DJ52_buk: same as comment #2, please define this enzyme name..

Point-by-point response to reviewers – Vogeli et al. 2022

We now define the enzyme name in Supplementary sheet 1 – Enzyme list and the Figure caption refers to it.

6. Figure 3 and also other figures: the colors used to represent different enzymes are very close and similar. The authors might want to revise the color code for a better differentiation of various enzymes..

We have adjusted the color code in all the figures to make it easier to visually identify which variant was used. All the reaction conditions, results and enzyme combinations are additionally listed for each figure in Supplementary sheet 1.

7. Line 284: delete “a”.

Deleted.

8. Line 305: For the *E. coli* work, what promoters/RBS are used.. Why did not further optimize the promoter/RBS usage to possibly further improve the product production levels?

The goal of the *in vivo* experiments with *E. coli* was to examine correlation with the *in vitro* platform. Since the *in vitro* platform evaluated different r-BOX enzymes at a fixed final concentration, we opted for using well-established expression vectors pETDuet-1 & pCDFDuet-1 (Novagen, Darmstadt, Germany) in which gene expression was controlled by a single promoter at a fixed inducer concentration. We share the reviewer’s interest in optimization of relative expression of r-BOX enzymes through different promoters/RBSs. We expect this to further improve pathway performance/product synthesis.

9. Along this line, in Dellomonaco et al. (Nature, 2011), the engineered *E. coli* strain “synthesized 2.2 g/L of n-butanol in 24 h at a yield of 0.28 g n-butanol per g total glucose consumed. When grown in a bioreactor using a higher concentration of glucose, this strain produced n-butanol at high titre of 14 g/L, yield (0.33 g n-butanol per g total glucose consumed) and rate (2 g n-butanol per g cell dry weight per h)”. While the butanol production level in *E. coli* in this study was much lower than this previously reported level even after extensive iPROBE-selection and optimization. How would the authors comment on this?

We appreciate this question and now comment on the observations. Notably, the focus of this work was on correlating the selectivity of the r-BOX pathway combinations between the three selected systems and for *E. coli* on the production of C6 acid and alcohol, which previous efforts have produced at a significantly lower titer. The 2011 paper was a top-down, systems-level approach in which we manipulated global regulators to activate all beta-oxidation genes. While this was very successful for butanol and longer chain fatty acids, the titer and selectivity toward medium chain products (C6-C10) was poor due to the fact we were not able to select a given enzyme for each step. Our current study demonstrates a rapid and high throughout way of selecting these enzymes that allowed us to selectively produce out target product hexanol and the highest titers reported to date (in flasks for that matter, not a controlled reactor with higher concentrations of carbon source). The previous work additionally suggests that further strain optimization in a similar top-down fashion for the new C6 producing strains might further improve the here reported yields and rates.

10. Line 339: For the work in *C. autoethanogenum*, why the acid production in *C. autoethanogenum* was not attempted here, or it has been attempted but the results were not good (and thus not presented)?.

The native AOR enzymes of *C. autoethanogenum* reduce the produced fatty acids to the corresponding alcohols. We did include thioesterase termination enzymes (those produced the acids both *in vitro* and in *E. coli*) in the screen for C4- and C6-alcohol production. We have added the following sentence to the main text to elaborate on this:

Point-by-point response to reviewers – Vogeli et al. 2022

Line 349: Note that the native aldehyde ferredoxin oxidoreductase (Ca_aor)²³ reduces the produced acids to the corresponding alcohols. Thioesterase termination enzymes were therefore included in screening for strains that produce C4- and C6-alcohols.

11. Line 348: Pfer, Pwl: please define.

The promoters used are defined in the cited reference 11 (our original iPROBE study) (as well as other recent work: <https://doi.org/10.1093/synbio/ysaa019>; <https://www.nature.com/articles/s41587-021-01195-w>) but we added some additional text to provide a reader with more context (see underlined text below).

“We next tested whether our iPROBE-selected r-BOX pathway variants also improved specificity for C4- and C6-alcohols in autotrophic, anaerobic *C. autoethanogenum*. In previous work¹⁹, we have established a modular vector system that allows us to build up *Clostridium in vivo* expression constructs directly from cell-free vectors by combining with different parts such as promoters of different strength. Note that the native aldehyde ferredoxin oxidoreductase (Ca_aor)²³ reduces the produced acids to the corresponding alcohols. Thioesterase termination enzymes were therefore included in screening for strains that produce C4- and C6-alcohols. Knowing the importance of the thiolase and the *C. autoethanogenum* native more efficient bcd-ETF system that replaces Td_ter in r-BOX¹¹ (previously shown to increase titers for butanol production), we first tested impact of expression of the iPROBE-selected thiolase (Ck_thIA) and the Bcd-Etf system in *C. autoethanogenum*¹¹ while keeping the rest of the core enzymes (i.e., Ck_hbd, Ca_crt, Ec_tesA) constant. For this we used the promoters already onboarded to our modular vector system, a *C. autoethanogenum* ferredoxin promoter (Pfer or Pfdx), a *C. autoethanogenum* phosphotransacetylase promoter (Ppta), a *C. autoethanogenum* pyruvate:ferredoxin oxidoreductase promoter (Ppfor) and *C. autoethanogenum* Wood-Ljungdahl cluster operon (Pwl or PacaA)¹⁹. The strain containing Ck_thIA under control of Pwl produced the highest titers of butanol and hexanol suggesting a fully active r-BOX cycle (**Figure 5A**).”

12. Fig. 5B: “Cf_buk” in the symbol (color code), but was actually not used in the figure. The symbol was added to the figure.

13. Line 424: Aor: please define.
AOR is now defined and cited.

14. Line 473: “the acetone and isopropanol gas fermentation process discussed here” ??
Changed to: a for-profit company pursuing commercialization of the *C. autoethanogenum* gas fermentation process discussed here.

Reviewer #2 (Remarks to the Author):

In this study, an iProbe rapid screening system was established in vitro, and verified in *Escherichia coli* and *C. autoethanogenum*, achieving very good results, which is very enlightening for the previous research. But a few additions are needed.

We thank the reviewer for highlighting that our paper has good results and is enlightening.

1. According to the description of the article, the reviewer understands that iProbe system is an application mode of cell-free protein expression system. May I ask if this understanding is correct? It is hoped that the author can draw a schematic diagram and briefly describe the system in the article.

Point-by-point response to reviewers – Vogeli et al. 2022

You are correct. iPROBE combines cell-free protein expression with the combinatorial assembly and prototyping of metabolic pathways. This helps to select the best pathways already in vitro and informs in vivo implementation. A scheme was added to the Supplementary Figure 1.

2.Line 142. What's the PANOx-SP CFPS system?Hope the author can make brief explanation to this abbreviation in the article.

PANOx-SP CFPS describes the components that are added to the cell-free protein synthesis mix used in this manuscript. The details are described in the referenced paper and in the method section of this manuscript. We additionally have cited the original paper: <http://jewettlab.northwestern.edu/wp-content/uploads/2021/04/3.pdf>

3.Line 154. Acetyl-coa synthesis pathway is regulated by strict feedback inhibition. Can the inactivation of acetyl-CoA consumption pathway increase acetyl-CoA concentration?Is there a documentary basis?

We did not measure acetyl-CoA concentration themselves, but knockout strains that prevent acetyl-CoA consumption pathways did show improved r-BOX product synthesis both in vivo (prior work by Kim et al. and Cheong et al.) and in vitro (Supplementary Figure 2).

4.Why do you choose *C. Autoethanogenum* as one of the validation platforms?

C. autoethanogenum was chosen as a target platform due to its industrial relevance of autotrophic growth on syngas from diverse sources and therefore the ability to produce the target molecule hexanol at a negative carbon balance. *C. autoethanogenum* is currently used at industrial scale (50,000 metric tons per year) for ethanol production from emissions from steel mill and ferroalloy production and over the past decades genetic tools have been developed that enabled efficient production of acetone or isopropanol (<https://www.nature.com/articles/s41587-021-01195-w>). In addition to that, it also provides a different metabolism to *E. coli* and we were interested to see how transferrable the results were across platforms.

Reviewers' Comments:

Reviewer #1:

Remarks to the Author:

This reviewer is primarily satisfied with the response from the authors. Some further comments as below:

1. To original Comment #2 (and #5): The reviewer appreciates that the authors have now connected the enzyme names with the 'Enzyme list' in the Supplementary data. However, can you add additional information in the 'Enzyme list' in the Supplementary data to identify the source (which microorganism, maybe the species name—for example, Ck_thIA, is this from *Clostridium kluyveri*?) of each enzyme? Maybe using the strain ID out of the 272 ABE strains collection as presented in the Nature Biotech paper?
2. Along these lines, the reviewer appreciates the efforts from the authors that they identified candidate genes from the large collection of *Clostridium* strains. However, some well known strains, particularly, the syngas fermenting *Clostridium carboxidivorans* P7 is known to naturally produce hexanoic acid and hexanol. Considering that *C. autoethanogenum* is also a syngas fermenting strain, if the relevant enzymes (for hexanoic acid and hexanol production) from *Clostridium carboxidivorans* P7 have not been included in the screening, the authors might have missed the 'best' candidate enzymes..
3. To original Comment #10: Could the identified enzymes be tested in a AOR negative *C. autoethanogenum* strain for hexanoic acid production?
4. Line 136: "hydroxybutyrate dehydrogenase – HBD" is the specific enzyme for C4 cycle. The 'general' enzyme names for the r-BOX should be used here.. Please also add these enzyme names into the caption description of Fig. 1A.

Reviewer #2:

Remarks to the Author:

The author answered my questions, and there were no more.

Point-by-point response to reviewers 2 – Vogeli et al. 2022

Responses are highlighted in blue

Reviewer #1 (Remarks to the Author):

This reviewer is primarily satisfied with the response from the authors. Some further comments as below:

1. To original Comment #2 (and #5): The reviewer appreciates that the authors have now connected the enzyme names with the 'Enzyme list' in the Supplementary data. However, can you add additional information in the 'Enzyme list' in the Supplementary data to identify the source (which microorganism, maybe the species name—for example, Ck_thlA, is this from *Clostridium kluyveri*?) of each enzyme? Maybe using the strain ID out of the 272 ABE strains collection as presented in the Nature Biotech paper?

We thank the reviewer for this question and have now added the organism from which the enzymes come from to the enzyme list in the Source Data sheet.

2. Along these lines, the reviewer appreciates the efforts from the authors that they identified candidate genes from the large collection of *Clostridium* strains. However, some well known strains, particularly, the syngas fermenting *Clostridium carboxidivorans* P7 is known to naturally produce hexanoic acid and hexanol. Considering that *C. autoethanogenum* is also a syngas fermenting strain, if the relevant enzymes (for hexanoic acid and hexanol production) from *Clostridium carboxidivorans* P7 have not been included in the screening, the authors might have missed the 'best' candidate enzymes..

We share the reviewer's enthusiasm for continuing to test additional enzymes and will look to do so in a future study. We have added the following sentence to the section where we already discuss the findings in *C. carboxidivorans* P7 (Line 384):

"While our study explored numerous enzyme sequences, further exploration of enzymes from *C. carboxidivorans* P7 may provide further benefits."

3. To original Comment #10: Could the identified enzymes be tested in a AOR negative *C. autoethanogenum* strain for hexanoic acid production?

We agree with the reviewer that the enzymes could be tested in such a strain and plan to explore this in the future.

4. Line 136: "hydroxybutyrate dehydrogenase – HBD" is the specific enzyme for C4 cycle. The 'general' enzyme names for the r-BOX should be used here.. Please also add these enzyme names into the caption description of Fig. 1A.

We thank the reviewer for raising this point and have changed to the use of hydroxyacyl-CoA dehydrogenase in Line 136. We additionally now name the general r-BOX enzymes in the caption of Fig. 1A.